# The DAMIC-M Experiment: Status and First Results

I. Arnquist[1], N. Avalos[2], D. Baxter[3,†], X. Bertou[2], N. Castelló-Mor[4], A.E. Chavarria[5],
N.J. Corso[3], J. Cortabitarte Gutiérrez[4], J. Cuevas-Zepeda[3], A. Dastgheibi-Fard[6], C. De
Dominicis[7], O. Deligny[8], J. Duarte-Campderros[4], E. Estrada[2], N. Gadola[9], R. Gaïor[10],
T. Hossbach[1], L. Khalil[10], B. Kilminster[9], A. Lantero-Barreda[4], I. Lawson[11], S. Lee[9],
A. Letessier-Selvon[10], P. Loaiza[8], A. Lopez-Virto[4], A. Matalon[3,10], K. McGuire[5], P. Mitra[5],
S. Munagavalasa[3], D. Norcini[3,*], G. Papadopoulos[10], S. Paul[3], A. Piers[5], P. Privitera[3,10],
P. Robmann[9], M. Settimo[7], R. Smida[3], M. Traina[10], G. Warot[6], I. Vila[4], R. Vilar[4], R. Yajur[3],
J-P. Zopounidis[10]

**1** Pacific Northwest National Laboratory (PNNL), Richland, WA, United States
**2** Centro Atómico Bariloche and Instituto Balseiro, Comisión Nacional de Energía Atómica
(CNEA), Consejo Nacional de Investigaciones Científicas y Técnicas (CONICET), Universidad
Nacional de Cuyo (UNCUYO), San Carlos de Bariloche, Argentina
**3** Kavli Institute for Cosmological Physics and The Enrico Fermi Institute, The University of
Chicago, Chicago, IL, United States
**4** Instituto de Física de Cantabria (IFCA), CSIC - Universidad de Cantabria, Santander, Spain
**5** Center for Experimental Nuclear Physics and Astrophysics, University of Washington,
Seattle, WA, United States
**6** LPSC LSM, CNRS/IN2P3, Université Grenoble-Alpes, Grenoble, France
**7** SUBATECH, Nantes Université, IMT Atlantique, CNRS-IN2P3, Nantes, France
**8** CNRS/IN2P3, IJCLab, Université Paris-Saclay, Orsay, France
**9** Universität Zürich Physik Institut, Zürich, Switzerland
**10** Laboratoire de physique nucléaire et des hautes énergies (LPNHE), Sorbonne Université,
Université Paris Cité, CNRS/IN2P3, Paris, France
**11** SNOLAB, Lively, ON, Canada
**(DAMIC-M Collaboration)**

*dnorcini@kicp.uchicago.edu

October 7, 2022

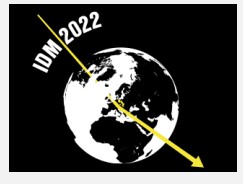

## Abstract

**The DAMIC-M (DArk Matter In CCDs at Modane) experiment employs thick, fully depleted silicon charged-coupled devices (CCDs) to search for dark matter particles with a target exposure of 1 kg-year. A novel skipper readout implemented in the CCDs provides single electron resolution through multiple non-destructive measurements of the individual pixel charge, pushing the detection threshold to the eV-scale. DAMIC-M will advance by several orders of magnitude the exploration of the dark matter particle hypothesis, in particular of candidates pertaining to the so-called "hidden sector." A prototype, the**

---

†Now at Fermi National Accelerator Laboratory, Batavia, IL, United States

Low Background Chamber (LBC), with 20g of low background Skipper CCDs, has been recently installed at Laboratoire Souterrain de Modane and is currently taking data. We will report the status of the DAMIC-M experiment and first results obtained with LBC commissioning data.

## Contents

## 1   The DAMIC-M experiment

The DAMIC-M (DArk Matter in CCDs at Modane) experiment [1] will use thick silicon charge-coupled devices (CCDs) to search for dark matter particles at the Laboratoire Souterrain de Modane in France (LSM). To be sensitive to nuclear and electronic recoils from scattering of light dark matter candidates (eV to GeV), the target is to achieve a low background rate of ∼0.1 dru and a 1 kg-year exposure. Additionally, the CCDs will operate with 2-3 electron ionization thresholds (∼5 eV) to meet the required sensitivity.

DAMIC-M is currently in the development phase, with anticipated installation in 2023-2024. The collaboration has accomplished many milestones thus far in preparation for the production of CCDs, detector design, as well as various CCD measurements. In particular, we have defined the strategy for producing silicon wafers with low cosmogenic activation and radon contamination by limiting the time above ground during fabrication, transport, and storage. We have also developed low background packaging procedures to ensure the CCD will be surrounded by clean materials. Figure 1 shows a prototype DAMIC-M four-CCD module packaged at the University of Washington, as well as the design for the 200-CCD array that will be contained in the electro-formed copper cryostat. To protect further from backgrounds, the array will be shielded by an innermost layer of ancient lead, inside a larger lead and polyethylene shield.

Several campaigns to characterize the performance, detector response, and background capabilities have been undertaken in the last few years. Measurements of Compton scattering on silicon down to 23 eV were presented by R. Smida at IDM2022 and have been accepted by Phys. Rev. D [2]. With the same experimental set-up at The University of Chicago, we are now measuring the nuclear recoil ionization efficiency to even lower thresholds. Crucially, single electron resolution with these large format, thick skipper CCDs has also been demonstrated in test chambers across multiple institutions. More details will be discussed in Section 2. Due to the excellent performance of the CCDs, we have recently installed a prototype detector, the Low Background Chamber (LBC), on-site at LSM. Through this effort, the collaboration has

gained enormous experience in working underground with the laboratory staff, handling low-background materials, and optimizing the operation parameters of the CCDs. Results from the first dark matter-electron scattering search with the LBC are presented in Section 3.

In anticipation of detector construction, DAMIC-M continues to make progress on the fabrication of low-background parts (including newly developed flex cables in collaboration with PNNL [3]), development of new CCD controller electronics, and evaluating the performance of DAMIC-M prototype CCDs.

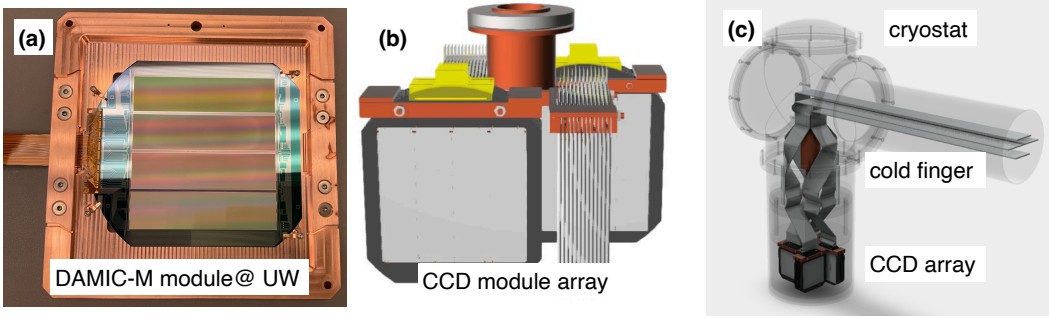

Figure 1: (a) DAMIC-M module with four CCDs in package at the University of Washington. (c) Design of CCD array with CCDs (light gray) on silicon pitch adaptor module (dark gray) in copper package (orange). (c) Model of one design option for the DAMIC-M cryostat, illustrating the thermal path, cabling, and CCD array.

## 2 Skipper CCDs for dark matter detection

DAMIC-M will use 200-massive (∼3.5 g), 9 Mpixel skipper CCDs in an array to achieve a kg-scale target mass. These devices feature a three-phase polysilicon gate structure with a buried p-channel, a pixel size of $15{\times}15\,\mu\mathrm{m}^2$, and a thickness of $670\,\mu\mathrm{m}$. The bulk of the devices is high-resistivity (10–20 kΩcm) n-type silicon, allowing for full-depletion at substrate biases ≥40 V. The CCDs were developed at Lawrence Berkeley National Laboratory MicroSystems Lab [4–6] and fabricated by Teledyne DALSA Semiconductor.

As in conventional CCDs, particle interactions in the silicon bulk generate charge carriers proportional to the energy deposition. A voltage bias applied between the bottom and top surfaces drifts the charge towards the pixel array in the z-direction. Simultaneously, as the charge drifts it also spreads with a Gaussian profile in the lateral direction due to thermal diffusion proportional to the drift length. Thus, the size of pixel clusters in the images allow for 3D reconstruction of interactions [7]. As the topology of the energy deposit depends on the particle type, CCDs can efficiently identify particles for background rejection.

Voltage clocks move the charge row-by-row towards the serial register and are then clocked to either end where a charge-to-voltage amplifier reads out the signal. An illustration of this process is shown in Figure 2(a). Skipper CCDs have special amplifiers (DAMIC-M will use $47/6\,\mu\mathrm{m}^2$ skipper amplifiers) that can make multiple non-destructive charge measurements (NDCMs) [8–10]. This type of floating gate amplifier allows the charge from one pixel to be moved back and forth across a measurement node $N_{skip}$ times before charge destruction. The measurements can then be averaged, and since they are uncorrelated, the readout noise is significantly reduced to $\sigma_{N_{skip}} = \sigma_1/\sqrt{N_{skip}}$, where $\sigma_1$ is the single-sample readout noise (i.e. the standard deviation of a single charge measurement). When $N_{skip}$ is large, we can reach sub-electron noise thresholds and be sensitive to single ionization signals. Figure 2(b) demonstrates the ability to measure single electrons (the individual Gaussian populations represent

$0e^-$, $1e^-$, $2e^-$ peaks) with a resolution of a fraction of an electron. Furthermore, the ability to identify single electrons provides an inherent energy conversion calibration.

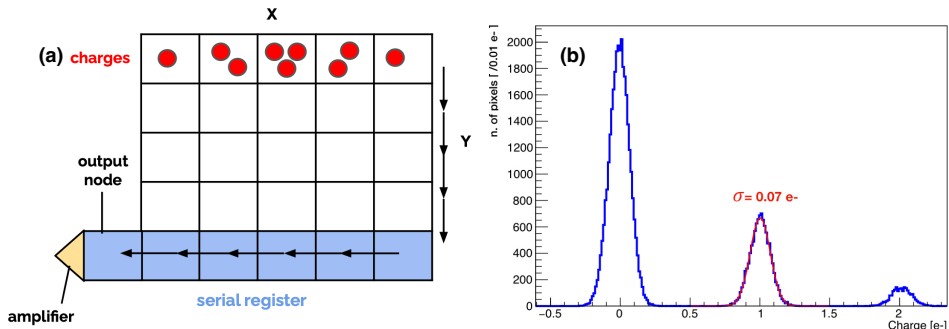

Figure 2: (a) Cartoon of how charge is moved in CCDs for readout. (b) Example pixel charge distribution demonstrating the single electron resolution capability of skipper CCDs. Each peak corresponds to the number of electrons detected.

Combining the information gained from the high spatial and energy resolution of skipper CCDs, the DAMIC-M detector can achieve a low background rate and detector threshold with excellent sensitivity to both nuclear and electronic recoils from light, GeV-scale Weakly Interacting Massive Particles (WIMPs) and sub-GeV dark-sector candidates, respectively. Figure 3 shows the projected sensitivity of DAMIC-M with a 1 kg-year exposure in searches for the absorption of hidden photons and dark matter-electron scattering via a heavy and light mediator.

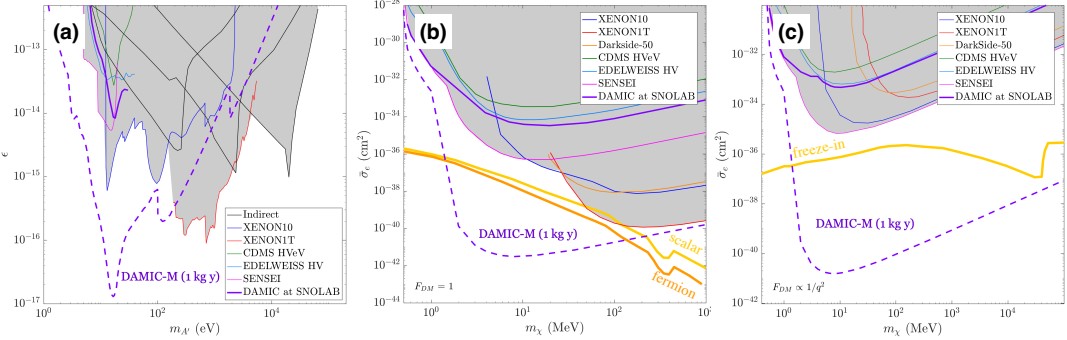

Figure 3: DAMIC-M sensitivity projections with 1 kg-year exposure for various dark matter candidates (a) hidden photons (b) heavy mediators (c) light mediators.

# 3   First results with the Low Background Chamber

To demonstrate the feasibility of the skipper CCD technology in a low-background environment, the DAMIC-M collaboration installed the LBC prototype at LSM at the end of 2021. The detector has reached a background level of $\mathcal{O}(10\,\mathrm{dru})$[1] and we have validated the design of various detector components and subsystems. Over the last months, the two 6k×4k large format skipper CCDs have operated stably inside the copper cryostat, shown in Figure 4(a). The air-driven cryocooler has maintained the CCDs at a temperature of ∼130 K and the pressure in-

---

[1]Future LBC upgrades will include replacing the standard copper CCD package with electro-formed copper, which we expect will reduce the background level to a few dru.

side the cryostat is held at $\sim 10^{-7}$ mbar by the turbo pump. The CCDs, data acquisition system, and instrumentation all run remotely and are monitored by the slow control system [11, 12]. During this time, no time-varying environmental backgrounds have been observed.

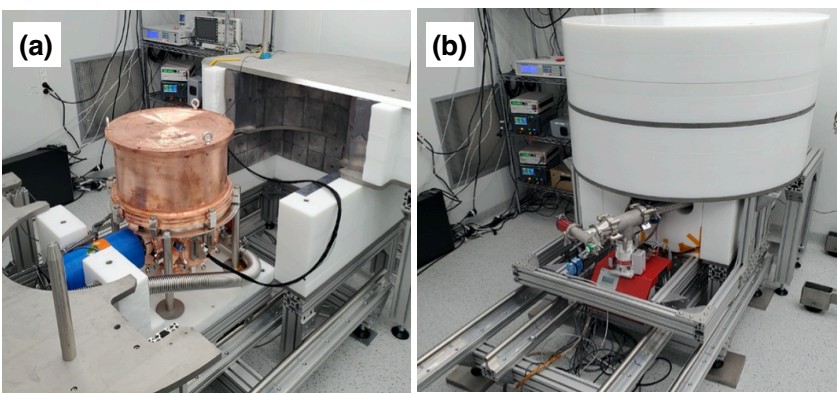

Figure 4: The LBC detector in the cleanroom at LSM with (a) internal lead shield only and (b) internal lead plus external lead and polyethylene shield. The external shielding provides a factor of 30 reduction in background rate.

Each of the two CCDs is read out by two amplifiers, producing a total of four images during each exposure. This allows for faster readout, and along with pixel binning, reduces the overall readout noise. The CCD operating parameters were optimized for collecting science data, resulting in an observed average resolution of $\sigma_{N_{skip}} \approx 0.2 e^-$ ($<1$ eV) at $N_{skip}$=650 and a dark current of $\sim 3 \times 10^{-3} e^-$/pixel/day. The measured dark current is a factor of 10 higher than anticipated (compared with DAMIC@SNOLAB [13]). We are working to understand if this is from an unidentified instrumental effect, stresses induced in the silicon, a light leak in the detector, or otherwise.

Multiple data sets have been acquired for commissioning and science purposes. To verify the performance of the detector and optimize the CCD parameters, the LBC was operated with an internal lead shield only, as shown in Figure 4(a). We were able to confirm the calibration strategy, develop analysis techniques, and reduce dark current with thermal tests during these runs. A 300 dru background level was achieved in this configuration. For science data the full shielding was assembled, see Figure 4(b), and the background reduces to $\sim$10 dru. From May to July, the LBC provided a 115 g-day exposure, from which we performed our first dark matter-electron scattering search.

Data analysis was performed by a set of well-vetted event selection steps and cross-checked across multiple analysis frameworks. First, images are selected by eliminating those that have a high dark current. Then we group pixel hits together using a clustering reconstruction algorithm, removing clusters with total energy $>10 e^-$ to distinguish higher energy events. Because we read out the CCDs with $10 \times 10$ binning to collect all the charge from one interaction in one pixel, this analysis only requires the identification of single-pixel events. A mask is applied to remove any multi-pixel clusters with residual charge resulting from charge transfer inefficiency (CTI). Additionally, cross-talk, or pixels with high charge observed in both amplifiers on the same CCD, are eliminated. The last step is to exclude defects in the silicon bulk by identifying "hot" columns that have a charge of $>2\sigma_{DC}$, where $\sigma_{DC}$ is the standard deviation of the dark current distrbution across all columns. An example single pixel charge-distribution (PCD) for one amplifier is shown in Figure 5.

To set the dark-matter electron limit, QEdark [14] is used to generate the differential rate of the dark matter signal with respect to energy. Halo parameters from Ref. [15] are used, as suggested by the dark matter community. The detector response is applied to the simu-

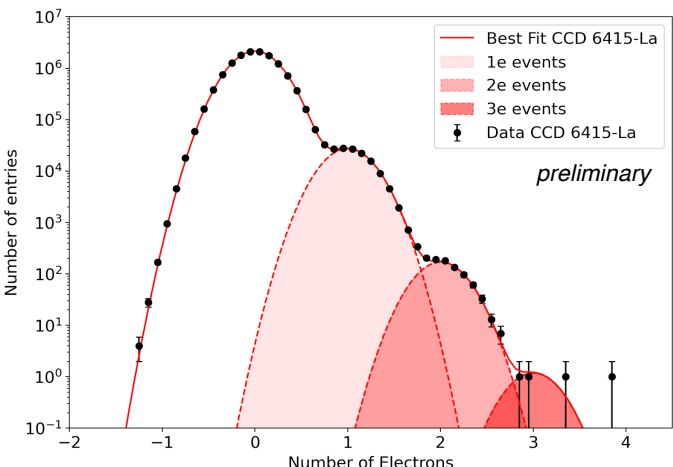

Figure 5: Single pixel-charge distribution from one amplifier in one LBC CCD.

lated signal to produce a probability distribution function. The response model includes the conversion of energy to charge for low energy ionization yields [16] and diffusion parameters measured with the LBC CCDs. The measured PCD in each amplifier is assumed to have a Poisson background with a Gaussian noise resolution. A global fit and binned joint likelihood minimization is performed to set the 90% C.L. upper limits in cross section-dark matter mass parameter space. The limits for the heavy and light mediator candidates are shown Figure 6.

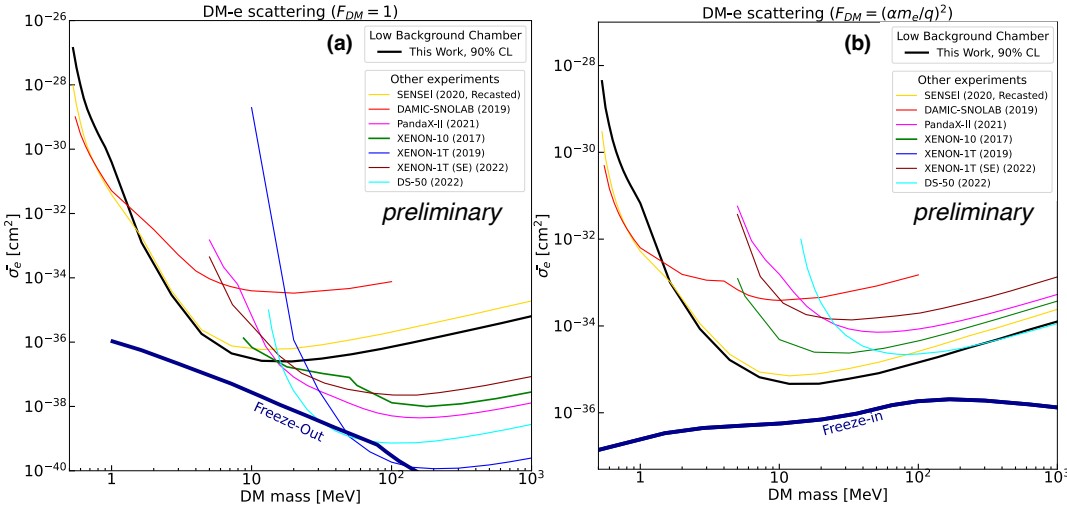

Figure 6: Preliminary DAMIC-M LBC 90% C.L. exclusions with 115 g-day exposure for (a) heavy mediator and (b) light mediator dark matter candidates.

We note here that, as illustrated in Figure 5, that the analysis yields a $4e^-$ event in one amplifier in this analysis. The probability of such an excess from the skipper readout noise model is ~15%. The interpretation of this is currently under investigation with an enhanced dataset, where we expect to improve the low-mass sensitivity for a world-leading limit.

# 4   Conclusion

DAMIC-M is a novel experiment using skipper CCDs to achieve low-energy thresholds that enable the search for light dark matter. The experiment is in the development phase towards

building a kg-scale CCD array housed within an extremely low background environment at LSM. The LBC has been installed since the end of 2021 and we have taken data to assess the background strategy and performance of the skipper CCDs. With a preliminary analysis, DAMIC-M has derived its first dark matter-electron limit with a 115 g-day exposure. Since the IDM conference, new data have been taken with reduced dark current and an improved, world-leading limit is forthcoming.

## Acknowledgements

The DAMIC-M project has received funding from the European Research Council (ERC) under the European Union's Horizon 2020 research and innovation programme Grant Agreement No. 788137, and from NSF through Grant No. NSF PHY-1812654. The work at University of Chicago and University of Washington was supported through Grant No. NSF PHY-2110585. This work was supported by the Kavli Institute for Cosmological Physics at the University of Chicago through an endowment from the Kavli Foundation. We also thank the College of Arts and Sciences at UW for contributing the first CCDs to the DAMIC-M project. IFCA was supported by project PID2019-109829GB-I00 funded by MCIN/ AEI /10.13039/501100011033.The Centro Atómico Bariloche group is supported by ANPCyT grant PICT-2018-03069. The University of Zürich was supported by the Swiss National Science Foundation.The CCD development work at Lawrence Berkeley National Laboratory MicroSystems Lab was supported in part by the Director, Office of Science, of the U.S. Department of Energy under Contract No. DE-AC02-05CH11231. D.N. is supported through the Kavli Institute for Cosmological Physics Fellowship and the Grainger Fellowship at The University of Chicago.

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
