# Peer review of "The DAMIC-M Experiment: Status and First Results"

_SciPost Physics Proceedings_

## Round 2 · Referee Report · Anonymous · 2022-11-17

Report

The paper presents preliminary results from the DAMIC-M experiment and derives limits on the dark matter-electron cross section for point-like and light-mediator induced interactions.

The presentation on the DAMIC-M part is sound and clear. However, there is a complete disregard of the many paralleling activities in the field. For example Fig. 3 and 5 show many experimental and theoretical results, but no reference is given to honor those contributions, not even to a community report. It is vexing to see how an entire collaboration can sign off on that. This should be fixed before I can recommend publication.

  • validity: high
  • significance: high
  • originality: high
  • clarity: good
  • formatting: good
  • grammar: excellent

Author:  Danielle Norcini  on 2022-11-17  [id 3040]

(in reply to Report 1 on 2022-11-17)
Category:
question
answer to question

Thank you for pointing this out. It was certainly an oversight on our part.

Would it be sufficient to provide a community report where these results can be found? Or should we provided references for all of the limits represented on Fig 3 and 5? Thank you.

Anonymous on 2022-11-18  [id 3047]

(in reply to Danielle Norcini on 2022-11-17 [id 3040])

Well, there is no rule, of course and it is up to the authors. But since they ask: a compass can be to honor other (original) contributions as one would like to see oneself cited.

---

## Editorial Decision

resubmitted